# School-based high-intensity interval training programs in children and adolescents: A systematic review and meta-analysis

Stephanie L. Duncombe[1,2]*, Alan R. Barker[2], Bert Bond[2], Renae Earle[1], Jo Varley-Campbell[3], Dimitris Vlachopoulos[2], Jacqueline L. Walker[1], Kathryn L. Weston[4], Michalis Stylianou[1]

**1** School of Human Movement and Nutrition Sciences, University of Queensland, Saint Lucia, Queensland, Australia, **2** Children's Health and Exercise Research Centre, Sport and Health Sciences, College of Life and Environmental Sciences, University of Exeter, Exeter, United Kingdom, **3** Department of Clinical, Educational and Health Psychology, University College London, London, United Kingdom, **4** School of Applied Sciences, Edinburgh Napier University, Edinburgh, United Kingdom

\* s.duncombe@uq.net.au

## Abstract

### Purpose

1) To investigate the effectiveness of school-based high-intensity interval training (HIIT) interventions in promoting health outcomes of children and adolescents compared with either a control group or other exercise modality; and 2) to explore the intervention characteristics and process outcomes of published school-based HIIT interventions.

### Methods

We searched Medline, Embase, CINAHL, SPORTDiscus, and Web of Science from inception until 31 March 2021. Studies were eligible if 1) participants aged 5–17 years old; 2) a HIIT intervention within a school setting $\geq$ 2 weeks duration; 3) a control or comparative exercise group; 4) health-related, cognitive, physical activity, nutrition, or program evaluation outcomes; and 5) original research published in English. We conducted meta-analyses between HIIT and control groups for all outcomes with $\geq$ 4 studies and meta-regressions for all outcomes with $\geq$ 10 studies. We narratively synthesised results between HIIT and comparative exercise groups.

### Results

Fifty-four papers met eligibility criteria, encompassing 42 unique studies (35 randomised controlled trials; 36 with a high risk of bias). Meta-analyses indicated significant improvements in waist circumference (mean difference (MD) = -2.5cm), body fat percentage (MD = -1.7%), body mass index (standardised mean difference (SMD) = -1.0), cardiorespiratory fitness (SMD = +1.0), resting heart rate (MD = -5bpm), homeostatic model assessment–insulin resistance (MD = -0.7), and low-density lipoprotein cholesterol (SMD = -0.9) for HIIT

**Data Availability Statement:** All relevant data are within a Supporting Information file (S4).

**Funding:** The author(s) received no specific funding for this work.

**Competing interests:** The authors have declared that no competing interests exist.

compared to the control group. Our narrative synthesis indicated mixed findings between HIIT and other comparative exercise groups.

## Conclusion

School-based HIIT is effective for improving several health outcomes. Future research should address the paucity of information on physical activity and nutrition outcomes and focus on the integration and long-term effectiveness of HIIT interventions within school settings.

## Trial registration number

PROSPERO CRD42018117567.

## Introduction

Recent evidence suggests that vigorous physical activity, as opposed to moderate physical activity, could be driving health benefits, such as reduced cardiometabolic risk, in youth [1–3]. Consequently, there has been an interest in high-intensity interval training (HIIT), defined as short bouts of vigorous exercise followed by recovery periods [4], as a potential method to acquire vigorous physical activity. For example, recent physical activity guidelines have called for research evaluating the effectiveness of HIIT [5, 6]. Available reviews in this area have demonstrated that HIIT can promote favourable changes in cardiometabolic risk, cardiorespiratory fitness (CRF), cognition and wellbeing in youth [7–15]. However, these reviews are confounded by the inclusion of studies conducted within different paediatric groups (e.g., athletic, or clinical populations) and in various settings (e.g., laboratory, school, clinical, and sports settings), introducing heterogeneity [9, 10, 16].

HIIT interventions conducted in the school setting need to be evaluated independently. Schools are an ideal setting for physical activity promotion as they can help reach a large percentage of children and adolescents with their policies and practices, existing infrastructure, and personnel who are or can be trained to support physical actvity [17]. Additionally, school-based interventions have the potential to be scalable and tend to be low cost [18]. However, this setting presents unique challenges, including time constraints, curriculum demands, and teacher workload and training [19]. Previous school-based physical activity interventions have had limited success at increasing physical activity levels [20–23], suggesting that novel approaches and improved delivery are necessary. HIIT may be a promising approach to use in schools given it aligns to habitual physical activity patterns in youth and the intermittent style of most modern sports [24, 25]. It is also associated with greater post-exercise enjoyment than continuous exercise and does not elicit unpleasant feelings [26]. Two recent reviews focused on HIIT in schools [7, 27]; however, recommendations for informing policy advocate for a systematic review with a meta-analysis [28]. Delgado-Floody *et al.* did conduct a meta-analysis but only focused on HIIT delivered in physical education classes in a population classified as overweight or obese, leading to the inclusion of only six studies [27]. Further, both reviews focused solely on cardiometabolic and fitness outcomes and did not consider outcomes related to psychological wellbeing, learning, nutrition, or program feasibility and sustainability [7, 27]. It is important to assess these outcomes to understand the uptake and sustainability of HIIT programs within the school setting.

Therefore, the objectives of this systematic review were to: 1) investigate the effectiveness of school-based HIIT interventions in promoting physical health, cognitive health, and psychological wellbeing of children and adolescents (5–17 years old); and 2) explore the intervention characteristics and process outcomes of published school-based HIIT interventions.

## Methods

This review follows the Preferred Reporting Items for Systematic Reviews and Meta Analyses (PRISMA) guidelines and was registered with the International Prospective Registry of Systematic Reviews (registration number CRD42018117567).

### Search strategy

We conducted a structured electronic search from inception until March 2021 via MEDLINE, EMBASE, CINAHL, SPORTDiscus, and Web of Science using subject headings and keywords related to "high intensity interval training", "high intensity interval exercise", "sprint interval training", "children", and "adolescents" (S1 File). These terms were selected based on relevant papers and a participant, intervention, comparison, and outcome (PICO) statement [29]. They were trialled and refined with the support of a librarian. Using forward citation chasing, we scanned reference lists of included full-text articles and systematic reviews for additional articles.

### Study selection and inclusion and exclusion criteria

After duplicate removal through Endnote (Clarivate Analytics, Philadelphia, USA) and Covidence software (Veritas Health Innovation, Melbourne Australia), titles and abstracts and subsequently full-text articles were screened independently by two reviewers. Discrepancies were resolved with a third reviewer. Articles were considered eligible for inclusion if they: 1) included 5–17-year-olds; 2) examined a HIIT intervention that occurred within a school setting at any point in the school day or before or after school; 3) had a minimum duration of two weeks; 4) had a control or a comparative exercise group; 5) examined outcomes related to health, cognition, physical activity, nutrition, or program evaluation; and 6) were original research articles published in English in peer-reviewed journals. Both randomised control trials (RCTs) and quasi-experimental studies were included as randomisation is not always feasible in school-based studies and informative literature could have been missed if only RCTs were included. We excluded studies if they focused on a specific disease or condition, or the youth athlete. Articles on children classified as obese or overweight were included. We placed no restrictions on the type of activity, intervention frequency, or cut-off intensity for "high-intensity", if an interval component was included. However, interventions had to be defined as "high-intensity" by the original authors. We attempted to contact authors when information was missing. If authors did not reply within two months, articles were excluded.

### Data extraction

Data extraction was conducted by one reviewer and verified by another. We extracted: 1) key characteristics about the study (study design, country), participants (inclusion/exclusion criteria, age, sex), and intervention (HIIT protocol and modality, adherence, attendance, location and time within the school, individual leading the intervention); 2) outcomes examined as specified in our protocol; and 3) results. For study results, we extracted the mean and standard deviation pre- and post-intervention for each group. When reported, we also extracted the mean difference, effect size, group significance, time significance, and group x time significance.

## Risk of bias and certainty of evidence

For our risk of bias assessment, we combined and adapted two tools recommended by the Cochrane Collaboration [29]. We used the Risk of Bias-2 (ROB-2) tool, which is designed for randomised studies, and for non-randomised quasi-experimental studies, we included a section of the Risk of Bias in Non-Randomised Studies (ROBINS-I) tool. For missing data, we used a cut-off of 15% based on quality assessments of other exercise interventions [30]. We modified the risk of bias due to deviations from the intended intervention section to appropriately reflect targeted interventions by evaluating adherence (attendance), adverse events, and program fidelity (meeting the desired exercise intensity). Each category received a bias score of "low", "some concerns", or "high". Overall bias was determined using the ROB-2 algorithm. Each study was assessed independently by two reviewers and discrepancies were resolved with a third reviewer. The certainty of evidence for each outcome included in a meta-analysis was assessed using the approach proposed by the Grading of Recommendations, Assessment, Development and Evaluation (GRADE) working group [31]. The evidence was classified into one of four levels of certainty: "high", "moderate", "low", or "very low". The certainty of the evidence was downgraded due to a high risk of bias, inconsistency within the results (unexplained heterogeneity), indirectness of the findings (lack of generalisability and/or external validity), imprecision (small sample sizes and/or wide confidence intervals) or detected publication bias. The certainty of evidence was upgraded for large effect sizes or if all plausible bias would reduce the determined effect size.

## Data synthesis and meta-analyses

For comparisons between the HIIT and control groups, we conducted meta-analyses for outcomes included in four or more studies and narratively synthesised the results for remaining outcomes that were reported in more than one study. For comparisons between HIIT and other exercise groups, we narratively synthesised available results reported in more than one study due to the heterogeneity among comparative group protocols.

Meta-analyses were conducted in R (Version 3.6.2; The R Foundation for Statistical Computing, Vienna, Austria) using the "meta" package. As this review included both randomised and quasi-experimental studies, we used change scores to analyse the effect of HIIT compared with control groups. When change score standard deviations were not reported, they were calculated from standard errors or confidence intervals, or imputed from correlation coefficients derived from other studies [32]. Random effect models were used to allow for variations between studies. For variables with measurements reported on multiple scales, a standardised mean difference (SMD) with inverse proportion weighting was used. For all other variables, the mean difference (MD) was used. Alpha was set at 0.05. We calculated heterogeneity using the $I^2$ statistic, with values between 0% to 40%, 30% to 60%, 50% to 90% and 75% to 100% representing trivial, moderate, substantial and considerable heterogeneity, respectively [29]. We used funnel plots to visually assess publication bias and Egger's test to quantify asymmetry and determine significance [33, 34].

We conducted meta-regressions and sub-analyses on unadjusted data to determine if the effects of the intervention differed due to intervention characteristics, including: 1) HIIT volume (minutes), defined as the total time performing HIIT including recovery periods but excluding warmup and cooldown, and 2) study duration (weeks). Additionally, meta-regressions were conducted on several participant characteristics: 1) mean age (years); 2) weight status classification (overweight and obese); and 3) sex (percentage of females). We removed the six studies where this percentage was not reported. Lastly, meta-regressions were conducted to understand the effect of study design and bias as follows: 1) RCTs vs quasi-experimental

studies; 2) high, some concerns, or low risk of overall bias; and 3) high, some concerns, or low bias due to deviations from the intended intervention. These sensitivity analyses were only completed for meta-analyses with an $n > 10$ to ensure there was adequate power and to limit false positives [35]. Alpha was set at 0.05 for moderator effects and only significant moderators are reported.

# Results

## Study characteristics

Fifty-four articles [32, 36–88] were eligible for inclusion in the review (Fig 1), consisting of 42 unique studies after combining the papers by Buchan et al. [46, 47], Costigan et al. [50–52], Cvetković et al. [53, 54], Arariza and Ruiz-Ariza et al. [39, 83], Van Biljon et al. [85, 86], Mucci et al. and Nourry et al. [76, 78], Lambrick et al. and McNarry et al. [63, 73], FIT-First study papers [56, 64] and Burn2Learn study papers [61, 65, 66, 68]. Thirty-nine of 42 studies included a control group, 13 contained an additional comparative group. The majority of the comparative groups included continuous exercise, but two studies used football and two used moderate intensity intervals. Four studies contained two different HIIT protocol groups, of which one combined HIIT and nutritional counselling. Three studies included

**PRISMA 2020 flow diagram for new systematic reviews which included searches of databases, registers and other sources**

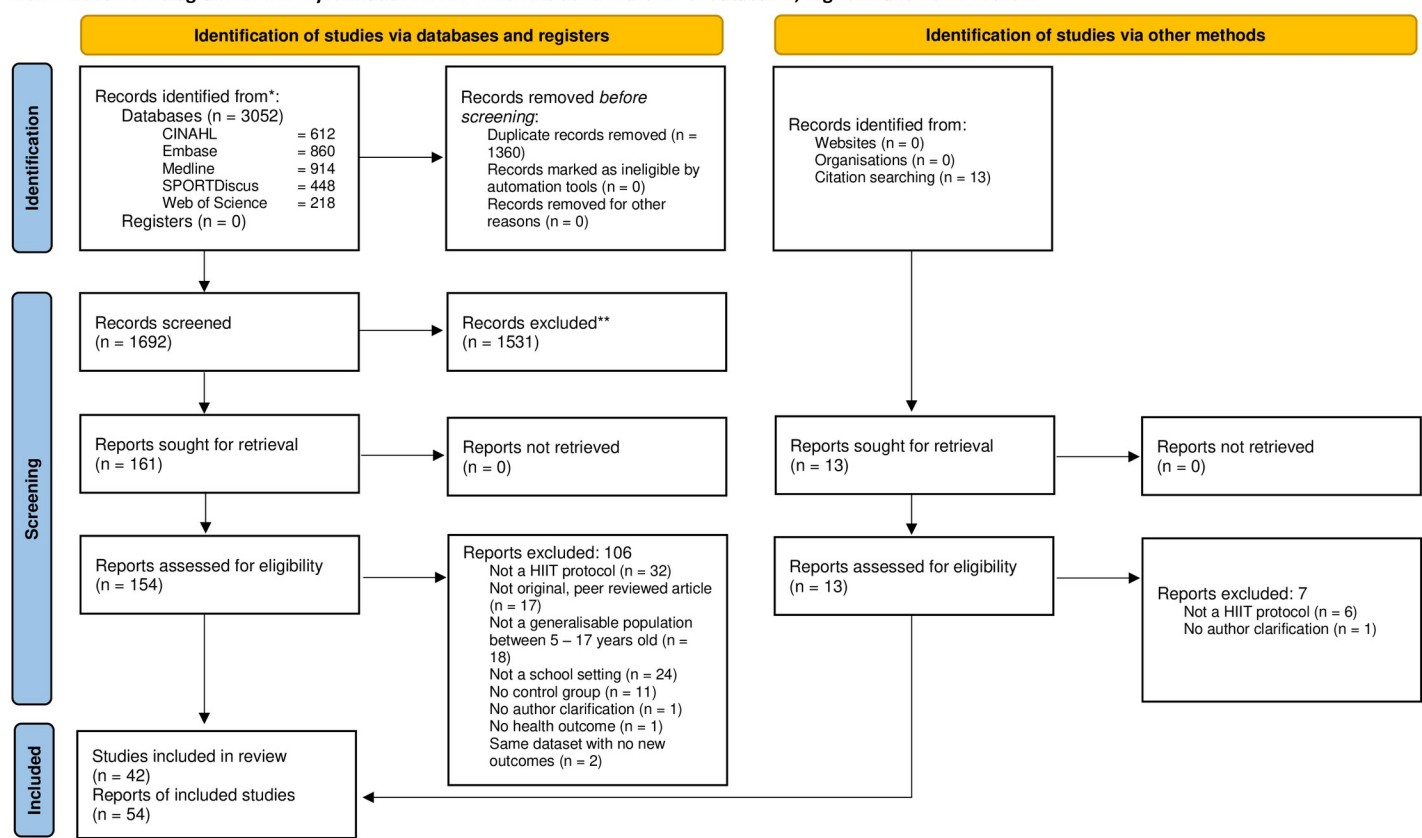

*Consider, if feasible to do so, reporting the number of records identified from each database or register searched (rather than the total number across all databases/registers).
**If automation tools were used, indicate how many records were excluded by a human and how many were excluded by automation tools.

*From:* Page MJ, McKenzie JE, Bossuyt PM, Boutron I, Hoffmann TC, Mulrow CD, et al. The PRISMA 2020 statement: an updated guideline for reporting systematic reviews. BMJ 2021;372:n71. doi: 10.1136/bmj.n71. For more information, visit: http://www.prisma-statement.org/

**Fig 1. Preferred Reporting Items for Systematic Reviews and Meta-Analyses (PRISMA) flow diagram.** HIIT = high-intensity interval training; WoS = Web of Science.

only a HIIT group with a comparative exercise group. Studies used a variety of modalities within their protocols, including running, cycling, dance, resistance training, circuits, games, strength training, and sports drills. The most common modality was running, and interval lengths within the interventions spanned from 10 seconds to a 4-minute bout of HIIT games. Summary study and HIIT program characteristics are reported in Table 1, with additional details available in Table 2.

**Table 1. Summary of study and program characteristics.**

| Characteristic | Category | N | % |
|---|---|---|---|
| | Europe | 25 | 59.5 |
| | Africa | 6 | 14.3 |
| **Continent** | Australia/New Zealand | 4 | 9.5 |
| | Asia | 4 | 9.5 |
| | South America | 3 | 7.1 |
| **Study Design** | Randomised | 35 | 83.3 |
| | Non-randomised | 7 | 16.7 |
| **Sex** | Male and Female | 22 | 52.3 |
| | Males only | 7 | 16.7 |
| | Females only | 8 | 19.1 |
| | Not Reported | 5 | 11.9 |
| **Sample Size** | <100 | 30 | 71.4 |
| | > 100 | 12 | 28.6 |
| **Intervention Length** | 2–7 weeks | 13 | 30.9 |
| | 8–12 weeks | 23 | 54.8 |
| | > 12 weeks | 6 | 14.3 |
| **Intervention Timing** | Before or after school | 4 | 9.5 |
| | During school hours | 7 | 16.7 |
| | During PE | 24 | 57.1 |
| | Not reported | 7 | 16.7 |
| **Intervention Frequency** | 1–2 times/week | 11 | 26.2 |
| | 3 times/week | 28 | 66.7 |
| | 4–5 times/week | 3 | 7.1 |
| **Intervention Facilitator** | External Trainers | 5 | 11.9 |
| | Researchers | 6 | 14.3 |
| | PE teachers | 7 | 16.7 |
| | Researchers and PE teachers | 4 | 9.5 |
| | Not Reported | 20 | 47.6 |
| **Intensity Results Reported** | Heart Rate | 20 | 47.6 |
| | Rating of Perceived Exertion | 1 | 2.4 |
| | Percentage of one repetition maximum | 1 | 2.4 |
| | Not reported | 20 | 47.6 |
| **Adverse Events** | Yes | 2 (2 students) | 4.8 |
| | No | 16 (969 students) | 38.1 |
| | Not reported | 24 | 57.1 |
| **Attendance Reported** | Yes | 15 | 35.7 |
| | No | 27 | 64.3 |

$N$ = number of studies; PE = physical education; % = the percentage of studies ($N$ / 42) with rounding completed to the nearest 10th.

**Table 2. Study characteristics.**

| Author (Year) Location, Study Design | Sample Size, Age, ˆSex Ratio (Girls/Boys) [a] | HIIT: Duration, Modality, Frequency, Total Volume of HIIT, [b] Bout Summary and Intensity | Comparative Exercise Group: Duration, Modality, Frequency, Total Volume of Exercise, [b] Bout Summary and Intensity | Control Group: Protocol Summary |
|---|---|---|---|---|
| Abassi et al. (2020), [36] Tunisia, RCT | 24, 16.5 ± 1.1, 100.0 / 0.0 | 12 weeks, Running, 3 x week, 900 minutes 6–8 x (30/30)) @ 100–100% MAV | 12 weeks, Running, 3 x week, 900 minutes 2 x (6 to 8 x (30/30)) @ 70–80% MAV | Told to maintain daily living |
| Adeniran et al. (1988), [37] Nigeria, RCT | 76, 15.6 ± 1.4, 100.0 / 0.0 | 8 weeks, Running, 3 x week, 768 minutes 4 x (240/240) @ > 90% HR Max | 8 weeks, Continuous Running 3 x week 576 minutes 3 miles (@ ≈ 8 min/mile) @ 80–85% HR Max | Not Recorded |
| Alonso-Fernandez et al. (2019), [38] Spain, RCT | 28, 15–16, 46.4 / 53.6 | 7 weeks, Body Weight Exercises, 2 x week, 92 minutes 8 x (20/10) @ NR | NA | Attended regular PE class |
| Arariza (2018)/ Ruiz Arirza et al. (2019) [39, 83] Spain, RCT | 184, 13.7 ± 1.3, 46.7 / 53.3 | 12 weeks, Circuit Exercises, 2 x week, 408 minutes 4 x (20/40) or (25/35) or (30/30) or (35/25) or (40/20) @ > 85% HR Max | NA | Static Stretching |
| Baquet et al. (2001), [41] France, Non-RCT | 551, 13.0 ± 1.0, 47.4 / 52.6 | 10 weeks, Running, 3 x week, 306 minutes 10 x (10/10) @ 100–120% MAV | NA | Attended regular PE class |
| Baquet et al. (2002), [40] France, Non-RCT | 53, 9.9 ± 0.4, 56.6 / 43.4 | 7 weeks, Running, 2 x week, 420 minutes 10 x (10/10) or 5 x (20/20) @ 100–130% MAV | NA | Attended regular PE class |
| Baquet et al. (2004), [43] France, RCT | 100, 9.8 ± 0.6, 54.0 / 46.0 | 7 weeks, Running, 2 x week, 420 minutes 10 x (10/10) or 5 x (20/20) @ 110–130% MAV | NA | Attended regular PE class |
| Baquet et al. (2010), [42] France, RCT | 72, 9.8 ± 1.2, 47.2 / 52.8 | 7 weeks, Running, 3 x week, 492 minutes 10 x (10/10) or 5 x (20/20) or 5 x (15/15) or 10 x (15/10) or 5 x (30/30) @ 100–130% MAV | 7 weeks, Continuous Running, 3 x week, 446 minutes 6 to 20 minutes @ 80–85% MAV | Attended regular PE class |
| Ben-Zeev et al. (2020), Israel, RCT | 40, 12–13 0.0 / 100.0 | 12 weeks, Running and resistance training, 3 x week, 720 minutes, 2 x (30s aerobic / 30s resistance) @ NR | NA | Attended regular PE class |
| Boddy et al. (2010), [44] England, RCT | 72, 9.8 ± 1.2, 47.2 / 52.8 | 3 weeks, Dance, 4 x week, 90 minutes 6 x (30/45) @ NR | NA | Not Reported |
| Bogataj et al. (2020), [45] Serbia, RCT | 66 15.7 ± 0.6 100.0 / 0.0 | 8 weeks, Body weight exercises 3 x week, + nutritionist 2 x week 360 minutes, 10 x (30s/15s) @ 80% Max HR | | Attended regular PE class |

(*Continued*)

**Table 2.** (*Continued*)

| Author (Year) Location, Study Design | Sample Size, Age, ˆSex Ratio (Girls/Boys) [a] | HIIT: Duration, Modality, Frequency, Total Volume of HIIT, [b] / Bout Summary and Intensity | | Comparative Exercise Group: Duration, Modality, Frequency, Total Volume of Exercise, [b] / Bout Summary and Intensity | | Control Group: Protocol Summary |
|---|---|---|---|---|---|---|
| Buchan et al. (2011), [46, 47] Scotland, Non-RCT | 47, 16.3 ± 0.5, 21.2 / 78.8 | 7 weeks, Running, 3 x week, 105 minutes, 4/5/6 x (30/30) or 6 x (30/20) @ NR | | 7 weeks, Continuous Running, 3 x week, 700 minutes, 20 minutes @ 70% VO₂ | | Attended regular PE class |
| Camacho-Cardenosa et al. (2016), [48] Spain, RCT | 47, 16.3 ± 0.5, 21.2 / 78.8 | 8 weeks, Running, 3 x week, 125 minutes, 3/4/5/6 x (20/60) or 4/5/6 x (20/40) or 4 x (20/20) @ NR | | 8 weeks, Continuous Running, 3 x week, 125 minutes, Equivalent time to HIIT workout @ 65–75% HR Max | | NA |
| Cheunsiri et al. (2018), [49] Thailand, RCT | 48, 11.0 ± 0.3, 0.0 / 100.0 | 12 weeks, Cycling, 3 x week, 864 minutes, 8 x (120/60) @ > 90% peak power output | | 12 weeks, Cycling, 3 x week, 144 minutes, 8 x (20/10) @ > 170% peak power output | | Told to maintain daily living |
| Costigan et al. (2015/ 2016/2018), [50–52] Australia, RCT | 65, 15.6 ± 0.6, 30.8 / 69.2 | 8 weeks, Running, 3 x week (2 in PE, one at lunch), 213 minutes, 8/9/10 x (30/30) @ > 85% HR Max | | 8 weeks, HIIT Resistance Training, 3 x week (2 in PE, one at lunch), 213 minutes, 8/9/10 x (30/30) @ 85% HR Max | | Attended regular PE class |
| Cvetkovic et al. (2018), [53, 54] Serbia, RCT | 42, 11–13, 0.0 / 100.0 | 12 weeks, Running, 3 x week, 660 minutes, 5 x (10/10) or 8 x (15/15) or 10 x (20/20) @ 100% MAV | | 12 weeks, Football, 3 x week, 1080 minutes, 4 x 8 minutes of playing @ NR | | Not Reported |
| Delgado Floody et al. (2018), [55] Chile, Non-RCT | 197, 8.4 ± 1.2, 54.8 / 45.2 | 28 weeks, Running, Jumps, Throws, 2 x week, NR (≈ 1512 minutes), 2/3/4 x (30-60/30-60) @ 80–95% HR Max | | NA | | Attended regular PE class |
| Elbe et al. (2016)/ Larsen et al. (2017), [56, 64] Denmark, RCT | 300, 9.3 ± 0.4, 52.6 / 47.4 | 44 weeks, or Running, or 5 x week, or 2640 minutes, 8 x (60/30) @ NR | 44 weeks Strength and Games 3 x week, 5280 minutes 6–10 x (30/45) @ NR | 44 weeks, or Football, or 5 x week, 2640 minutes Continuous play | 44 weeks, Football, 3 x week, 5280 minutes Continuous play | Attended regular PE class |
| Espinoza-Silva et al. (2019), [57] Chile, Non-RCT | 274, 7–9, 56.2 / 43.8 | 28 weeks, Running, Jumps, Throws, 2 x week, NR (≈ 1960 minutes), NR x (30-60/60-120) and 3–4 x (240/60-120) @ 8–10 RPE | | NA | | Attended regular PE class |
| Gamelin et al. (2009), [58] France, RCT | 38, 9.6 ± 1.2, 50.0 / 50.0 | 7 weeks, Running, 3 x week, 492 minutes, 10 x (10/10) or 5 x (20/20) or 5 x (15/15) or 10 x (15/10) or 5 x (30/30) or 2o x (5/15) @ 100–130% MAV | | NA | | Not Recorded |
| Granacher et al. (2011), [59] Switzerland, RCT | 34, 8.6 ± 0.5, 43.8 / 56.2 | 10 weeks, Strength Training, 2 x week, 1400 minutes, 3 x (10–12 reps/180-240s) @ 70–80% 1 rep max | | NA | | Attended regular PE class |

(*Continued*)

**Table 2.** (Continued)

| Author (Year) Location, Study Design | Sample Size, Age, ˆSex Ratio (Girls/Boys) [a] | HIIT: Duration, Modality, Frequency, Total Volume of HIIT, [b] Bout Summary and Intensity | | | | | Comparative Exercise Group: Duration, Modality, Frequency, Total Volume of Exercise, [b] Bout Summary and Intensity | Control Group: Protocol Summary |
|---|---|---|---|---|---|---|---|---|
| Haghshenas et al. (2019) [60] Iran, RCT | | 8 weeks, | | | | | NA | Active walks in the school yard |
| | 100, | Running, | | | | | | |
| | 14.0 ± 1.0, | 3 x week, | | | | | | |
| | 0.0 / 100.0 | 430.5 minutes | | | | | | |
| | | 2–4 (60-120/240/300) | | | | | | |
| | | @ NR MAV | | | | | | |
| Ketelhut et al. (2020), [62] Germany, RCT | | 12 weeks, | | | | | NA | Attended regular PE class |
| | 46, | Games, Circuits, Choreographies | | | | | | |
| | 10.8 ± 0.6, | 2 x week, | | | | | | |
| | 45.7 / 54.3 | 480 minutes | | | | | | |
| | | 2–6 x (20-120/30-90) | | | | | | |
| | | @NR HR Max | | | | | | |
| Lambrick et al. (2016)/McNarry et al. (2015), [63, 73] England, RCT | | 6 weeks, | | | | | NA | Attended regular PE class |
| | 55, | Games | | | | | | |
| | 9.2 ± 0.8, | 2 x week, | | | | | | |
| | 45.5 / 54.5 | 408 minutes | | | | | | |
| | | 6 x (360/120) games and 4 min circuit | | | | | | |
| | | @> 85% HR Max | | | | | | |
| Logan et al. (2016), [63] New Zealand, RCT | | 8 weeks, | | | | | | NA |
| | 24, | Aerobic and Resistance | | | | | | |
| | 16.0 ± 1.0, | 3 x week (2 HIIT, 1 resistance), | | | | | | |
| | 0.0 / 100.0 | 173.3 minutes | 234.7 minutes | 296.0 minutes | 357.3 minutes | 418.7 minutes [c] | | |
| | | 1 x (4 x 20/10) | 2 x (4 x 20/10) | 3 x (4 x 20/10) | 4 x (4 x 20/10) | 5 x (4 x 20/10) | | |
| | | Resistance = 3 x 8–12 of 3 compound movements | | | | | | |
| | | @ 90–100% HR Max for HIIT and 70% 1RM for Resistance | | | | | | |
| Lubans et al. (2020)/Kennedy et al. (2020)/Leahy et al. (2019)/ Leahy et al. (2020), [61, 65, 66, 68] Australia, RCT | | 52 weeks, | | | | | NA | Attended regular PE class |
| | 670, | Aerobic, Resistance, Dance, Boxing | | | | | | |
| | 16.0 ± 0.4, | 3 x week (½ year: 2 in PE, one own time, ½ year: all own time), | | | | | | |
| | 44.6 / 55.4 | ≈ 1248 minutes (using 8 min average/session and 52 weeks) | | | | | | |
| | | 8–16 x (30/30) | | | | | | |
| | | @> 85% HR Max | | | | | | |
| Martin et al. (2015), [69] Scotland, RCT | | 7 weeks, | | | | | NA | Attended regular PE class |
| | 49, | Running, | | | | | | |
| | 16.9 ± 0.4, | 3 x week, | | | | | | |
| | 24.5 / 75.5 | 108 minutes | | | | | | |
| | | 4–6 x (30/30) | | | | | | |
| | | @ NR | | | | | | |
| Martin-Smith et al. (2018), [70] Scotland, RCT | | 4 weeks, | | | | | NA | Attended regular PE class |
| | 56, | Running, | | | | | | |
| | 17 ± 0.3 | 3 x week, | | | | | | |
| | 37.5 / 62.5 | 66 minutes | | | | | | |
| | | 5–6 x (30/30) | | | | | | |
| | | @ NR (used a sprint pacer) | | | | | | |
| McManus et al. (1997), [71] England, RCT | | 8 weeks, | | | | | 8 weeks, | Not Reported |
| | 45, | Running, | | | | | Continuous Cycling, | |
| | 9.6 ± 0.5 | 3 x week, | | | | | 3 x week, | |
| | 100.0 / 0.0 | 304 minutes | | | | | 320 minutes | |
| | | 3–6 x (10/30) and 3–6 x (30/90) | | | | | 20 minutes | |
| | | @ NR (used a distance) | | | | | @ 80–85% HR Max | |
| McManus et al. (2005), [72] Hong Kong, RCT | | 8 weeks, | | | | | 8 weeks, | Not Reported |
| | 45, | Cycling, | | | | | Continuous Cycling, | |
| | 10.4 ± 0.5 | 3 x week, | | | | | 3 x week, | |
| | 0.0 / 100.0 | 320 minutes | | | | | 320 minutes | |
| | | 7 x (30/165) | | | | | 20 minutes | |
| | | @ Peak Power elicited during VO$_2$ test | | | | | @ 70–85% HR Max | |

(*Continued*)

**Table 2.** (Continued)

| Author (Year) Location, Study Design | Sample Size, Age, ˆSex Ratio (Girls/Boys) [a] | HIIT: Duration, Modality, Frequency, Total Volume of HIIT, [b] Bout Summary and Intensity | Comparative Exercise Group: Duration, Modality, Frequency, Total Volume of Exercise, [b] Bout Summary and Intensity | Control Group: Protocol Summary |
|---|---|---|---|---|
| McNarry et al. [d] (2020), [74] Wales, RCT | | 26 weeks, | NA | Not Reported |
| | 33, | Circuits and Games, | | |
| | 13.5 ± 0.8 | 3 x week, | | |
| | 45.4 / 55.6 | 1890 minutes | | |
| | | (10-30/10-30) | | |
| | | @ > 90% HR Max | | |
| Moreau et al. (2017), [75] New Zealand, RCT | | 6 weeks, | NA | Board Games |
| | 305, | Video Workouts, | | |
| | 9.9 ± 1.7 | 5 x week, | | |
| | 61.3 / 38.7 | 150 minutes | | |
| | | 1 x (20/20) and 1 x (20/30) and 1 x (20/40) and 1 x (20/50) and 1 x (20/60) | | |
| | | @ NR | | |
| Mucci et al. (2013)/ Nourry et al. (2005), [76, 78] France, RCT | | 8 weeks, | NA | Not Recorded |
| | 18, | Running, | | |
| | 10.0 ± 0.7 | 2 x week, | | |
| | 38.9 / 61.1 | 198 minutes | | |
| | | 10 x (10/10); 5 x (20/20); 5 x (15/15); 10 x (15/10); 5 x (30/30) | | |
| | | @ 100–130% MAV | | |
| Muntaner-Mas et al. (2017), [77] Spain, RCT | | 16 weeks, | NA | Attended regular PE class |
| | 80, | Circuit, | | |
| | 15.8 ± 0.5 | 2 x week, | | |
| | NR | 320 minutes | | |
| | | 10 x (45/15) | | |
| | | @ > 85% Max HR | | |
| Racil et al. (2013), [79] Tunisia, RCT | | 12 weeks, | 12 weeks, | Not Recorded |
| | 36, | Running, | Running | |
| | 15.9 ± 1.2 | 3 x week, | 3 x week, | |
| | 100.0 / 0.0 | 672 minutes | 672 minutes | |
| | | 6–8 x (30/30) | 6–8 x (30/30) | |
| | | @ 100–100% MAV and 50% MAV on rest | @ 70–80% MAV and 50% MAV on rest | |
| Racil et al. (2016a), [80] Tunisia, RCT | | 12 weeks, | 12 weeks, | Not Recorded |
| | 47, | Running, | Running | |
| | 14.2 ± 1.2 | 3 x week, | 3 x week, | |
| | 100.0 / 0.0 | 440 minutes | 440 minutes | |
| | | 4–8 x (15/15) | 4–8 x (15/15) | |
| | | @ 100 MAV and 50% MAV on rest | @ 80% MAV and 50% MAV on rest | |
| Racil et al. (2016b), [81] Tunisia, RCT | | 12 weeks, | 12 weeks, | Not Recorded |
| | 75, | Running, | Running and Plyometrics | |
| | 16.6 ± 0.9 | 3 x week, | 3 x week, | |
| | 100.0 / 0.0 | 672 minutes | 996 minutes | |
| | | 6–8 x (30/30) | 4 x (15/15) for plyometrics | |
| | | | 6–8 x (30/30) for sprints | |
| | | @ 100% MAV and 50% MAV on rest | @ 100% MAV and 50% MAV on rest | |
| Reyes Amigo et al. (2021), [82] Chile, RCT | | 11 weeks, | 11 weeks, | NA |
| | | HIIT Games, | Moderate Intensity Games, | |
| | 48, | 2 x week, | 2 x week, | |
| | 9.5 ± 0.5 | 510 minutes, | 510 minutes, | |
| | 66.7 / 33.3 | 4 x (6-minute intermittent game) | 4 x (6-minute continuous game) | |
| | | @75–95% Max HR or 6–8 / 10 RPE | @60–74% Max HR or 4–5 / 10 RPE | |
| Segovia et al. (2020), [84] Spain, RCT | | 6 weeks, | NA | Played Ringo |
| | 154 | Games and Circuit, | | In regular |
| | 10.7 ± 0.8 | 2–3 x week, | | PE class |
| | 47.4 / 52.6 | 195 minutes | | |
| | | 1 x 300–420 for games | | |
| | | 5–8 x (40/20) for circuit | | |
| | | @85–90% | | |

(*Continued*)

**Table 2.** (Continued)

| Author (Year) Location, Study Design | Sample Size, Age, ˆSex Ratio (Girls/Boys) [a] | HIIT: Duration, Modality, Frequency, Total Volume of HIIT, [b]; Bout Summary and Intensity | Comparative Exercise Group: Duration, Modality, Frequency, Total Volume of Exercise, [b]; Bout Summary and Intensity | Control Group: Protocol Summary |
|---|---|---|---|---|
| Van Biljon et al. (2018), [85, 86] South Africa, Non-RCT | 120, 11.1 ± 0.8, 61.4 / 38.6 | 5 weeks, Running, 3 x week, 337.5 minutes, 10 x (60/75) @ > 80% Max HR | 5 weeks, Walking, 3 x week, 495 minutes, 33 minutes @ 65–70% Max HR | 5 weeks, Alt. Running and Walking, 3 x week, 400.5 minutes, 3 weeks of sprints 2 weeks of walking — Not Recorded |
| Weston et al. (2016), [87] England, Non-RCT | 101, 14.1 ± 0.3, 37.6 / 62.4 | 10 weeks, Dance, Soccer, Boxing, Basketball, 3 x week (2 in PE, 1 after school/at lunch), 119.3 minutes, 4–7 x (45/90) @ >90% Max HR | NA | Attended regular PE class |
| Williams et al. (2000), [88] England, RCT | 45, 10.0 ± 0.2, 0.0 / 100.0 | 8 weeks, Running, 3 x week, 330 minutes, 3–6 x (10/30) and 3–6 x (30/90) @ 100% MAV and 50% MAV on rest | 8 weeks, Cycling, 3 x week, 420 minutes, 20 minutes @ 80–85% HR Max | Normal everyday activities |

Study characteristics including participant characteristics (sample size, age, sex ratio), protocol characteristics for HIIT and the comparative exercise group (duration–in weeks, modality–style of exercise, frequency–number of times per week, total time, and a general description with intensity), and protocol characteristics for the control group; HIIT = high intensity interval training; Max HR = maximum heart rate; MAV = maximal aerobic velocity; NA = not applicable; NR = not recorded; PE = physical education; RCT = randomised control trial; 1RM = 1 repetition maximum.

ˆ reported as mean and standard deviation (x ± x), or where not provided as range (x–x).

a reported as frequency (%).

b time in intervention excluding warm up and cool down.

c This study compares 5 different HIIT protocols with different volumes of HIIT.

d Data extracted only for healthy children.

## Process outcomes

Over half of the studies (24 of 42) were completed during physical education (PE) class but only 11 documented that PE teachers played a role in their delivery, while 20 studies did not provide information on the intervention facilitator. Attendance data was reported in only 35.7% of studies (Table 1). It varied across studies from 63% [65] to above 90% [32, 45, 59, 62, 63, 76, 79–82, 85]. Different intensity targets were set for participants in interventions. Four studies did not specify a target and instead used terminology such as "suitably high" and "sprint maximally" [32, 44, 47, 75]. For all other studies, a target threshold for heart rate, speed, power, or RPE was provided to participants. The lowest intensity target among any study was 75% of maximum heart rate during high intensity games with both work and rest included [82]. Assessment of whether these targets were achieved (fidelity) only occurred in 47.6% studies, with heart rate as the most commonly used tool. Session intensity was most often reported as an average heart rate across all participants and sessions. Five studies [48, 69, 70, 87, 88] used the average heart rate during only work intervals whereas other studies used an average that included both work and rest intervals or did not specify what was included. One study [64] reported the average time spent in different heart rate zones by participants and one study reported the number of students that achieved the desired heart rate during sessions in addition to the average and maximum heart rate [61]. Among the studies that reported session intensity, two studies did not use heart rate, with one using an RPE scale [55] and the other using a percentage of a one maximum repetition [59].

### Risk of bias and certainty of evidence

Thirty-six of the 42 studies had a "high" risk of bias (Table 3), mostly related to deviation from the intended intervention and missing data. High bias related to randomisation was noted least often. Four studies were classified as having "some concerns", and only two as having a "low" risk of bias. Using the GRADE approach, the certainty of the outcomes ranged between "very low" and "moderate" (S2 File). The most common reasons for downgrading the evidence were risk of bias and inconsistency within the findings. The certainty of evidence for body fat percentage, body mass index (BMI), low-density lipoprotein (LDL), and CRF was upgraded by one point due to large effect sizes within the findings.

### Physical health outcomes

Table 4 reports results for all outcomes examined in two or more studies comparing HIIT to a control group. Forest plots for all meta-analyses are presented in S3 File. HIIT was favoured in meta-analyses for waist circumference, body fat percentage, BMI, CRF, resting heart rate, homeostatic model assessment–insulin resistance (HOMA-IR), and LDL. Publication bias was significant for body fat percentage ($p = 0.049$), BMI ($p = 0.003$) and CRF ($p = 0.001$). According to the meta-regression results, having an entire population classified as overweight or obese significantly moderated the results for waist circumference ($n = 7$, $\beta$ = -0.56, $p = 0.009$), body fat percentage ($n = 9$, $\beta$ = -2.11, $p < 0.0001$), and BMI ($n = 9$, $\beta$ = -1.38, $p < 0.0001$), with a greater decrease noted in this population. Additionally, there was a greater increase in CRF in these studies ($n = 5$, $\beta$ = 1.01, $p = 0.007$). Having an entire population classified as overweight or obese also explained some of the heterogeneity present in the model for waist circumference (Residual heterogeneity: $I^2 = 36\%$, $p = 0.06$). Studies with a higher volume of HIIT were associated with a greater decrease in body fat percentage ($\beta$ = -0.002, $p < 0.0001$) and BMI ($\beta$ = -0.001, $p = 0.0014$). Studies with a longer protocol duration had a greater decrease in body fat percentage ($\beta$ = -0.12, $p = 0.0004$). Including a higher percentage of girls was also associated with a greater decrease in body fat percentage ($\beta$ = -0.01, $p = 0.0377$) and BMI ($\beta$ = -0.01, $p = 0.0109$). Studies with a high risk of bias due to deviations from the intended intervention had a significantly greater increase in CRF compared to studies with low bias ($\beta$ = 1.03, $p = 0.013$). When only the 5 studies with low bias were included in the analysis, heterogeneity was not significant ($I^2 = 14\%$, $p = 0.32$) and the random effects model was still significant (SMD = 0.41, 95% CI = 0.12 to 0.70) [47, 52, 63, 86, 88]. The method used to assess CRF (20 m shuttle run, cycle ergometer, or treadmill ergometer) and body fat percentage (Dual X-ray absorptiometry, bioelectrical impedance, or skinfold estimation) did not significantly moderate the results.

Table 5 reports findings for all outcomes examined in two or more studies comparing HIIT and comparative exercise groups, with no significant differences reported between the two groups for most health outcomes. Across all health outcomes, only three studies had results that favoured HIIT [79, 80, 85], while one study had results that favoured continuous exercise [47].

### Psychosocial and cognitive outcomes

As shown in Table 4, there were heterogeneous results for inhibition and memory when comparing HIIT and control groups in the four studies where these outcomes were examined. A variety of tests were used to investigate these two outcomes, with no two studies using the same battery of tests so no meta-analyses were performed. Two studies demonstrated no improvement to wellbeing after HIIT [51, 68], while one found an improvement in inactive

**Table 3. Risk of bias assessment based on ROB-2 and ROBINS.**

| | Randomised Control Trials | | | | | |
|---|---|---|---|---|---|---|
| | Randomisation and Selection Bias | Bias due to Missing Data | Measurement Bias | Bias due to Deviations from the Intended Intervention | Bias due to Analysis and Selection of Reported Results | Overall Risk of Bias |
| Abassi et al. (2020) [36] | Some Concerns | High | Some Concerns | High | High | High |
| Adeniran et al. (1988) [37] | Low | Low | Some Concerns | High | Some Concerns | High |
| Arariza (2018)/Ruiz Arirza et al. (2019) [39, 83] | Low | Low | Low | Low | Some Concerns | Some Concerns |
| Alonso-Fernandez et al. (2019) [38] | Low | High | Some Concerns | High | Some Concerns | High |
| Baquet et al. (2004) [43] | Low | Some Concerns | Some Concerns | High | Some Concerns | High |
| Baquet et al. (2010) [42] | Low | Some Concerns | High | High | High | High |
| Boddy et al. (2010) [44] | Some Concerns | Some Concerns | Low | Some Concerns | High | High |
| Ben-Zeev et al. (2020) [32] | High | Low | Some Concerns | High | Some Concerns | High |
| Bogataj et al. (2020) [45] | Some Concerns | Low | Low | High | Some Concerns | High |
| Buchan et al. (2011) [46, 47] | High | Low | Some Concerns | Low | High | High |
| Camacho-Cardenosa et al. (2016) [48] | Low | Low | Some Concerns | Some Concerns | Some Concerns | Some Concerns |
| Cheunsiri et al. (2018) [49] | Some Concerns | High | Some Concerns | High | Some Concerns | High |
| Costigan et al. (2015/2016/2018) [50–52] | Low | Low | Low | Low | Low | Low |
| Cvetkovic et al. (2018) [53, 54] | Some Concerns | High | Some Concerns | Some Concerns | Some Concerns | High |
| Elbe et al. (2016)/Larsen et al. (2015) [56, 64] | Low | High | Some Concerns | High | Low | High |
| Gamelin et al. (2009) [58] | Low | Some Concerns | Some Concerns | High | Some Concerns | High |
| Granacher et al. (2011) [59] | Low | Low | Some Concerns | Low | Some Concerns | Some Concerns |
| Haghshenas et al. (2019) [60] | Low | Low | Some Concerns | High | Some Concerns | High |
| Lambrick et al. (2016)/McNarry et al. (2015) [63, 73] | Some Concerns | Some Concerns | Some Concerns | Low | Some Concerns | High |
| Ketelhut et al. (2020) [62] | Low | High | Some Concerns | High | Some Concerns | High |
| Lubans et al. (2020)/Leahy et al. (2018)/ Leahy et al. (2020)/Kennedy et al. (2020) [62, 65, 66, 68] | Some Concerns | Low | Some Concerns | High | Low | High |
| Logan et al. (2016) [67] | High | Low | Some Concerns | Low | Some Concerns | High |
| Martin et al. (2015) [69] | Low | High | Some Concerns | High | Some Concerns | High |
| Martin-Smith et al. (2018) [70] | Low | Low | Some Concerns | High | Low | High |
| McManus et al. (1997) [71] | High | High | Some Concerns | High | Some Concerns | High |

(*Continued*)

**Table 3.** (Continued)

| | | | | | | |
|---|---|---|---|---|---|---|
| McManus et al. (2005) [72] | Some Concerns | High | Some Concerns | High | Some Concerns | High |
| McNarry et al. (2020) [74] | Low | High | Some Concerns | Some Concerns | Some Concerns | High |
| Moureau et al. (2017) [75] | Some Concerns | Low | Low | Low | Low | Low |
| Mucci et al. (2013)/Nourry et al. (2005) [76, 78] | Some Concerns | Some Concerns | Low | High | Some Concerns | High |
| Racil et al. (2013) [79] | Some Concerns | Low | Some Concerns | High | Some Concerns | High |
| Racil et al. (2016a) [80] | High | Low | Some Concerns | Some Concerns | Some Concerns | High |
| Racil et al. (2016b) [81] | Some Concerns | Low | Some Concerns | High | Some Concerns | High |
| Reyes Amigo et al. (2021) [82] | High | Low | Some Concerns | High | Some Concerns | High |
| Segovia et al. (2020) [84] | Low | High | Some Concerns | High | Some Concerns | High |
| Williams et al. (2000) [88] | Some Concerns | Low | Some Concerns | Low | High | High |
| | **Quasi-Experimental Studies** | | | | | |
| | Bias due to Confounding | Bias due to Missing Data | Measurement Bias | Bias due to Deviations from the Intended Intervention | Bias due to Analysis and Selection of Reported Results | Overall Risk of Bias |
| Baquet et al. (2001) [41] | Some Concerns | High | Some Concerns | High | Some Concerns | High |
| Baquet et al. (2002) [40] | Low | High | Some Concerns | Some Concerns | Some Concerns | High |
| Delgado Floody et al. (2018) [55] | High | High | Some Concerns | High | Some Concerns | High |
| Espinoza-Sliva et al. (2019) [57] | Low | High | Some Concerns | High | High | High |
| Muntaner-Mas et al. (2017) [77] | High | High | Some Concerns | High | High | High |
| Van Biljon et al. (2018) [85, 86] | High | Low | Some Concerns | Low | Some Concerns | High |
| Weston et al. (2016) [87] | Low | Low | Some Concerns | Some Concerns | Some Concerns | Some Concerns |

Risk of bias assessment for each study included in the review.; Bias due to missing data uses a 15% cut-off; Bias due to deviations from the intended intervention was modified to reflect an exercise intervention by assessing the fidelity of attaining high intensity, the attendance, the adverse events, and the qualifications of the person leading the intervention. ROB-2 = risk of bias; ROBINS = risk of bias in non-randomised studies; RCT = randomised control trial

children only [83]. No between-group difference was present for motivation levels towards completing the HIIT workouts [51, 68].

## HIIT intervention enjoyment

Enjoyment of HIIT was examined in four studies [49, 52, 56, 61]. Two [49, 56] used the validated Physical Activity Enjoyment Scale (PACES) questionnaire and determined that team sports elicited significantly greater enjoyment than individual sports [56], that 20-second bouts were enjoyed more than 120-second bouts [49], and that enjoyment was significantly associated with improvement in running performance [56]. Two studies [52, 61] used Likert questions to examine enjoyment alongside motivation, fatigue, and satisfaction, and found

**Table 4. Summary of outcomes between HIIT and control groups for all outcomes reported in ≥ 2 studies.**

| | Outcome | Participants (Studies) | Analysis | Certainty of the Evidence (GRADE) | Key Finding | Heterogeneity |
|---|---|---|---|---|---|---|
| **Body Composition** | Waist circumference | 1175 (14) | MA + MR | ⊕⊕⊕⊖ | Favoured HIIT, MD = -2.5 cm (-3.1 to -1.9) [36, 44, 52, 55, 57, 63, 70, 77, 79–81, 84, 85, 87] | $I^2$ = 47%, $p$ = 0.01 |
| | Body fat percentage | 1893 (19) | MA + MR | ⊕⊕⊖⊖ | Favoured HIIT, MD = -1.7% (-2.3 to -1.1) [36, 38, 40, 41, 43–45, 47, 49, 54, 55, 57, 63, 77, 79–81, 84, 87] | $I^2$ = 93%, $p$ < 0.01 |
| | Body Mass Index | 2450 (22) | MA + MR | ⊕⊕⊖⊖ | Favoured HIIT, SMD = -0.9 (-1.3 to -0.6) [36, 38, 41, 42, 44, 45, 47, 49, 52, 54, 55, 57, 63, 68, 69, 74, 77, 79–81, 85, 87] | $I^2$ = 92%, $p$ < 0.01 |
| | Muscle mass | 264 (5) | MA | ⊕⊕⊖⊖ | Summary statistic NS [45, 49, 54, 63, 87] | $I^2$ = 43%, $p$ = 0.12 |
| | Lean mass | 297 (4) | MA | ⊕⊖⊖⊖ | Summary statistic NS [44, 54, 64, 80] | $I^2$ = 90%, $p$ < 0.01 |
| | Hip circumference | 126 (3) | Narrative | | NS in 3 studies [44, 63, 70] | |
| | Bone density and content | 300 (2) | Narrative | | NS in 2 studies [44, 64] | |
| **Cardiovascular Health** | Systolic blood pressure | 872 (11) | MA | ⊕⊕⊕⊖ | Summary statistic NS [44, 47, 49, 54, 55, 57, 62, 70, 80, 85, 87] | $I^2$ = 29%, $p$ = 0.14 |
| | Diastolic blood pressure | 872 (11) | MA | ⊕⊕⊖⊖ | Summary statistic NS [44, 47, 49, 54, 55, 57, 62, 70, 80, 85, 87] | $I^2$ = 68%, $p$ < 0.01 |
| | Resting heart rate | 381 (6) | MA | ⊕⊕⊖⊖ | Favoured HIIT, MD = -5 bpm (-7 to -2) [49, 54, 55, 58, 80, 85] | $I^2$ = 52%, $p$ = 0.03 |
| | Heart rate variability | 147 (2) | Narrative | | Favoured HIIT in 1 study [86], NS in 1 study [58] | |
| | Aortic pulse wave velocity | 166 (2) | Narrative | | Favoured HIIT in 1 study [62], NS in 1 study [85] | |
| **Blood Profile** | Glucose | 447 (10) | MA | ⊕⊕⊕⊖ | Summary statistic NS [36, 47, 53, 69, 70, 79–81, 86, 87] | $I^2$ = 0%, $p$ = 0.81 |
| | Insulin | 321 (8) | MA | ⊕⊖⊖⊖ | Summary statistic NS [36, 47, 69, 70, 79–81, 85] | $I^2$ = 93%, $p$ < 0.01 |
| | HOMA-IR | 211 (5) | MA | ⊕⊕⊕⊖ | Favoured HIIT, MD = -0.7 (-1.1 to -0.4) [69, 70, 79–81] | $I^2$ = 95%, $p$ < 0.01 |
| | Triglycerides | 279 (6) | MA | ⊕⊖⊖⊖ | Summary statistic NS [47, 49, 54, 70, 79, 87] | $I^2$ = 84%, $p$ < 0.01 |
| | Total cholesterol | 279 (6) | MA | ⊕⊖⊖⊖ | Summary statistic NS [47, 49, 54, 70, 79, 87] | $I^2$ = 84%, $p$ < 0.01 |
| | High-density lipoprotein | 254 (5) | MA | ⊕⊖⊖⊖ | Summary statistic NS [47, 49, 54, 70, 79, 87] | $I^2$ = 36%, $p$ = 0.18 |
| | Low-density lipoprotein | 153 (4) | MA | ⊕⊕⊕⊖ | Favoured HIIT, SMD = -0.9 (-1.2 to -0.5) [47, 49, 70, 79] | $I^2$ = 0%, $p$ = 0.53 |
| | Leptin | 152 (3) | Narrative | | Favoured HIIT in 2 studies [80, 81], NS in 1 study [49] | |
| | Adiponectin | 206 (4) | Narrative | | Favoured HIIT in 3 studies [47, 79, 81], NS in 1 study [49] | |
| | C-reactive Protein | 265 (3) | Narrative | | Favoured HIIT in 1 study [85], NS in 2 studies [47, 87] | |
| **Aerobic & Muscular Fitness** | Cardiorespiratory fitness (all methods)** | 2099 (25) | MA + MR | ⊕⊖⊖⊖ | Favoured HIIT, SMD = 1.0 (0.7 to 1.3) [36, 38, 40–42, 44, 45, 47, 49, 52, 58, 63, 68–72, 74, 78–81, 86–88] | $I^2$ = 83%, $p$ < 0.01 |
| | Cardiorespiratory fitness (relative $VO_2$) † | 403 (11) | MA | ⊕⊕⊕⊖ | Favoured HIIT, MD = 3.1 ml/min/kg (2.4 to 3.8) [40, 42, 44, 49, 58, 63, 72, 76, 79, 81, 88] | $I^2$ = 50%, $p$ = 0.03 |
| | Cardiorespiratory fitness (shuttles) ‡ | 299 (5) | MA | ⊕⊖⊖⊖ | Favourite HIIT, MD = 10.4 shuttles (1.9 to 18.9) [46, 52, 65, 69, 87] | $I^2$ = 88%, $p$ < 0.01 |
| | Standing long jump | 1428 (5) | MA | ⊕⊕⊕⊖ | Summary statistic NS [41, 43, 52, 68, 77] | $I^2$ = 84%, $p$ < 0.01 |
| | Countermovement jump | 212 (5) | MA | ⊕⊕⊕⊖ | Summary statistic NS [45, 46, 53, 59, 81] | $I^2$ = 53%, $p$ = 0.07 |
| | Push ups | 735 (2) | Narrative | | Favoured HIIT in 1 study [68], NS in 1 study [52] | |
| | Handgrip Strength | 146 (2) | Narrative | | NS in 2 studies [45, 77] | |
| | Sit ups | 624 (2) | Narrative | | NS in 2 studies [41, 43] | |
| | Sprint time | 331 (3) | Narrative | | Favoured HIIT in 2 studies [46, 64], NS in 1 study [53] | |
| | Flexibility | 693 (3) | Narrative | | NS in 3 studies [41, 43, 53] | |
| | Balance | 334 (2) | Narrative | | NS in 2 studies [59, 64] | |

*(Continued)*

**Table 4.** (Continued)

| | Outcome | Participants (Studies) | Analysis | Certainty of the Evidence (GRADE) | Key Finding | Heterogeneity |
|---|---|---|---|---|---|---|
| Cognition and Wellbeing | Inhibition | 1199 (4) | Narrative | | Favoured HIIT in 3 studies [32, 39, 75], NS in 1 study [68] | |
| | Memory | 1199 (4) | Narrative | | Favoured HIIT in 2 studies [32, 75], NS in 2 studies [39, 68] | |
| | Wellbeing | 919 (3) | Narrative | | Favoured HIIT in 1 study [83], NS in 2 studies [51, 68] | |
| | Motivation levels | 126 (2) | Narrative | | NS in 2 studies [51, 68] | |
| Physical activity and Nutrition | Vigorous Physical Activity | 791 (3) | Narrative | | Favoured HIIT in 2 studies [50, 70], NS in 1 study [68] | |
| | Moderate Physical Activity | 791 (3) | Narrative | | Favoured HIIT in 1 study [70] NS in 2 studies [50, 68] | |
| | Moderate-to-Vigorous Physical Activity | 843 (3) | Narrative | | Favoured HIIT in 1 study [87] NS in 2 studies [44, 68] | |
| | Step Count | 790 (3) | Narrative | | Favoured HIIT in 1 study [68], NS in 2 studies [44, 49] | |
| | Caloric intake | 71 (3) | Narrative | | NS in 2 studies [69, 81] | |

Participants (studies) = number of participants (number of studies) included. HOMA-IR–homeostatic model assessment–insulin resistance; MA–meta-analysis; MR–meta-regressions; HIIT–high intensity interval training; NS–not significant; MD–mean difference; SMD–standardised mean difference.

** cardiorespiratory fitness was examined using either 20 m shuttle runs, cycle ergometer, or treadmill ergometer and it was reported either as the number of shuttles completed, or as $VO_2$, which was either measured by a metabolic cart or estimated using an equation. The type of measurement did not significantly moderate the results.

† Body mass relative maximum oxygen consumption directly assessed by metabolic cart.

‡ Number of shuttles completed in the 20 m shuttle run test using a mean difference.

that students and teachers were satisfied with the HIIT workouts, and the majority intended to continue using the workouts.

## Physical activity levels and energy intake

Five studies used accelerometers to quantify physical activity outcomes for HIIT and control groups [44, 50, 68, 70, 87], one used a pedometer [49], and one used the Physical Activity Questionnaire for Children [49]. Physical activity outcomes were reported using different outcome variables (Table 4), with no more than three studies reporting the same variable, therefore meta-analyses were not performed. Heterogeneous findings were present for physical activity variables and no significant differences existed between the HIIT and control groups for caloric intake in the two studies examining the outcome (Table 4).

## Comparing HIIT protocols

Four studies compared different HIIT protocols. Two compared aerobic training to aerobic training plus resistance or plyometric training [52, 81]. A third compared a shorter bout length of higher intensity to longer bouts of lower intensity [49], and the last looked at different doses of HIIT by changing the number of sets [67]. No clear effect of dose or bout length was found in these studies [49, 67] and heterogenous findings were reported when resistance training was added to aerobic training [52, 81].

## Discussion

This systematic review advances the findings of previous reviews [9–11] by investigating a broader range of outcomes associated with school-based HIIT interventions through comprehensive statistical analysis. The results of this review demonstrate that school-based HIIT is an effective strategy for improving various health outcomes compared with control groups. However, there are heterogenous findings when HIIT is compared to other exercise modalities.

**Table 5. Summary of outcomes between HIIT and comparative exercise groups for all outcomes reported in $\geq$ 2 studies.**

| | Outcome | Participants (studies) | General Finding |
|---|---|---|---|
| Body Composition | Waist circumference | 137 (4) | Favoured HIIT in 1 study [85], NS in 3 studies [36, 79, 80] |
| | Body fat percentage | 168 (6) | Favoured HIIT in 1 study [79], Favoured comparator in 1 study [47], NS in 4 studies [36, 48, 54, 80] |
| | BMI | 235 (7) | NS in 7 studies [36, 42, 47, 54, 79, 80, 85] |
| Cardiovascular Health | Systolic blood pressure | 145 (4) | Favoured HIIT in 1 study [86], NS in 3 studies [47, 54, 80] |
| | Diastolic blood pressure | 145 (4) | NS in 4 studies [47, 54, 80, 86] |
| | Resting heart rate | 112 (2) | Favoured HIIT in 1 study [86], NS in 1 study [54] |
| Blood Profile | Glucose | 191 (6) | NS in 6 studies [36, 47, 53, 79, 80, 85] |
| | Insulin | 170 (5) | Favoured HIIT in 2 studies [79, 80], Favoured comparator in 1 study [47], NS in 2 study [36, 85] |
| | HOMA-IR | 79 (3) | NS in 3 studies [36, 79, 80] |
| | Triglycerides | 76 (3) | Favoured HIIT in 1 study [79], NS in 2 studies [47, 54] |
| | Total cholesterol | 76 (3) | NS in 3 studies [47, 54, 79] |
| | High-density lipoprotein | 55 (2) | NS in 2 studies [47, 79] |
| | Low-density lipoprotein | 55 (2) | NS in 2 studies [47, 79] |
| Aerobic & Muscular Fitness | Cardiorespiratory fitness | 225 (7) | Favoured HIIT in 1 study [85], NS in 6 studies [36, 42, 71, 72, 79, 80, 88] |
| | Countermovement jump | 220 (2) | NS in 2 studies [46, 53] |

Participants (studies) = number of participants (number of studies) included. HOMA-IR = homeostatic model assessment–insulin resistance; HIIT = high intensity interval training; NS = not significant.

Overall, most studies had a high risk of bias, therefore the results need to be interpreted cautiously. Although findings support HIIT can be a useful tool within schools to promote a range of health benefits, they also highlight that further research is needed to examine the meaningful integration of these interventions within schools.

## Physical health outcomes: HIIT compared with control

Youths with obesity have an increased risk of developing cardiometabolic conditions [89–91], making it an important outcome to monitor. Improvements to body composition were documented across the included studies in this review with moderate (waist circumference, body fat percentage) or low (BMI) certainty according to GRADE when comparing HIIT with control groups. Our body fat percentage summary effect (1.7%) is similar to another meta-analysis on HIIT, where a 1.6% (95% CI: 0.5% to 2.9%) change was noted in favour of HIIT compared to a combination of non-training controls and moderate intensity groups [9]. While our summary effect for BMI differs to a systematic review on all school-based physical activity interventions that reported no significant change [20], it is equivalent to a previous meta-analysis ($n$ = 8) that compared HIIT to both control groups and moderate intensity comparative groups across various settings [9]. Our findings also have the potential to be clinically meaningful. For example, while we do not have individual data points in this synthesis, a summary effect demonstrating a decrease in waist circumference of 2.5 cm (1.9 to 3.1 cm) is equitable to a decrease from the 90th to 85th percentile in 16-year-old boys or a decrease from the 90th to 80th percentile in 7-year-old girls [91], but this could be influenced by baseline values. In our review, studies that only included students classified as overweight or obese had significantly greater health benefits as a result of HIIT. As increased adiposity is associated with future

disease related morbidity and mortality [92], decreasing adiposity, especially in populations classified as obese and overweight, is critical to prevent disease [93]. No significant differences were seen for lean mass, muscle mass, or hip circumference within our systematic review. However, this could be due to the smaller sample sizes for these outcomes.

We can say with moderate certainty that CRF is significantly improved as a result of HIIT interventions compared with a control group. The large effect size ($d$ = 0.9) established in this study mirrors that of two previous meta-analyses on HIIT ($d$ = 1.05 in adolescents and $d$ = 1.11 in adolescents classified as obese or overweight) [9, 13]. Relevant literature shows a positive association between vigorous activity and CRF, corroborating this finding [94]. According to our findings, there was an increase of 3.1 ml/kg/min (2.4 to 3.8 ml/kg/min) in the HIIT group after the intervention compared with the control group in the 11 studies that directly determined peak $\dot{V}O_2$, maximum oxygen consumption. This difference has the potential to be clinically meaningful as a lower CRF is associated with higher cardiometabolic risk in children, independent from physical activity and adiposity [95]. Further, children and adolescents in the lowest quartile for fitness have a greater risk for developing cardiovascular disease compared with those in the highest quartile for fitness [96]. Muscular fitness was examined in fewer studies than CRF, with no difference between the HIIT and control group noted for jumping, handgrip strength or sit-ups through meta-analyses and narrative synthesis. These will be important outcomes to study in more detail as HIIT protocols diversify and further involve different muscle groups. HIIT could have effects on muscular fitness with current research demonstrating a link between vigorous activity and a variety of muscular fitness test outcomes [97, 98].

The LDL and HOMA-IR blood biomarkers were significantly improved following HIIT compared with control groups in this review. However, the studies within these meta-analyses comprised of mainly populations classified as overweight or obese (50% and 60% of studies, respectively), which could be driving this change. The lack of change to other biomarkers for cardiometabolic health, including blood pressure, fasting glucose, triglycerides, and total cholesterol, could be reflective of the fact that baseline measures were within normal thresholds. We might expect to see changes for these variables in populations where the initial levels are elevated, such as in students who are classified as overweight or obese. This is consistent with findings from a recent review that demonstrated that while physical activity interventions in youths classified as obese are capable of producing favourable changes in biomarkers, the same dose is not effective for non-obese youths [99]. However, it is still important to encourage physical activity in all students regardless of their body composition as there is a strong positive association between total physical activity and blood biomarkers in youths [99] and puberty is a crucial period for the development of hypertension later in life [100].

### HIIT protocols and comparative exercise

More research is needed to determine if differences exist between HIIT and comparative exercise protocols in the school setting. Our narratively synthesised results did not detect any differences between HIIT and moderate continuous exercise or other comparative exercise protocols, such as moderate intensity intervals or football. However, HIIT provides educators with another option for promoting physical activity and has several unique characteristics that may make it effective in this setting. It can be short and simple to conduct, enabling it to be performed in a classroom setting [65, 101], while partly alleviating concerns that it will compete for time with curricular demands, which is a common reason compromising the effectiveness of school-based interventions [19].

## Process outcomes

Overall, process outcomes were documented poorly throughout these studies. The lack of fidelity and attendance data makes it difficult to assess if students received the intended HIIT intervention, which is critical as the intensity of exercise is likely to be important in driving physiological changes. Even for studies that stated that the desired intensity was achieved, this was most often based on an average heart rate across all participants and sessions, which does not allow provide readers with information on how many students successfully completed the intervention. Further, mean peak heart rate was occasionally reported as an outcome measure, which does not capture the variability within sessions. It will be important for future studies to appropriately document the attendance and fidelity of these interventions for proper evaluation [102]. This could help inform readers of HIIT protocols that are more likely to achieve high intensity in this setting. The intervention timing and facilitators varied between studies, and this could have implications on the reach, maintenance, and scalability of studies. However, the variation in the HIIT protocols across studies suggests that there are opportunities to tailor protocols to specific classes or students to appropriately engage and challenge them, and in turn optimise associated outcomes. There was no evidence of integration within the school curriculum in these studies, even though integration can mitigate the overloading teachers and provide staff with appropriate resources, which are shown to improve implementation [19] and should be a focus of future studies.

## Future directions

High-quality studies are needed in this area to be able to reach more robust conclusions as significant limitations were identified in the studies included in this review. Specifically, the lack of power calculations and documentation whether the intervention took places as was intended, along with the high levels of missing data that were unaccounted for in the analyses lead to studies with high risk of bias. Future studies should focus on 1) providing justification for their sample size; 2) reporting adherence, fidelity, and whether blinding occurred to determine deviations from the intended intervention; 3) and performing statistical analyses that account for any missing data.

The body of work focusing on school-based HIIT would benefit from additional studies examining cognitive, physical activity and nutrition outcomes. Our findings for cognitive outcomes are similar to those of a systematic review focusing on the impact of HIIT in adolescents across all settings that determined that HIIT may improve cognitive function but highlighted the need for more relevant studies [103]. These outcomes are important to assess, especially within the school setting, as they are related to academic success and improvements in this domain are likely to encourage schools to engage with HIIT [104]. Our narrative synthesis included heterogenous findings for the few studies that examined physical activity levels. More studies investigating physical activity levels and nutritional intake will be useful to help understand the impact of HIIT on these outcomes and whether incorporating HIIT leads to any compensatory behaviours in these domains, as recommended by a recent expert statement [105]. This expert statement also calls for further research into the benefits that are specific to students classified as overweight or obese [105]. Our meta-regressions demonstrated that studies including only those classified as overweight or obese moderated the results for waist circumference, body fat, BMI, and CRF. Moving forward, this will be important to also assess for other variables. As the body of evidence grows, it will be important to investigate potential sex and pubertal differences. Future studies should ensure that they report participants' pubertal stages in addition to their sex. Further, it will be important for future studies to report results stratified by sex

and maturity status to enable the effects of these variables to be understood. Additionally, beyond sex and maturity, studies should aim to investigate these health outcomes are present across schools in different contexts with varying physical activity policies and practices as these vary greatly between countries, school systems, and individual schools.

While this review supports the effectiveness of HIIT interventions in schools, factors related to their feasibility and maintenance must also be considered to improve meaningful short-term and long-term outcomes. It will be important to further investigate enjoyment and affect among HIIT protocols in schools to understand the likelihood for future engagement in these programs [106]. Current research on HIIT has displayed favourable results on enjoyment during and after exercise compared to moderate-intensity continuous training [107]. One strategy to facilitate high levels of student enjoyment may be involving students in the design of HIIT protocols. Affording students ownership in the design of HIIT protocols has the additional potential to also enhance students' accountability, participation, confidence and perceived competence in completing the workouts when the interventions reach the implementation phase [108]. This may be particularly useful for girls given they are less likely to enjoy school physical education and have on average a lower self-perceived physical ability [109]. Beyond students, studies should consider engaging other key stakeholders (e.g., teachers, parents, principals, local policy makers) in designing the intervention to increase the likelihood that interventions are maintained. Co-designing relevant interventions with teachers and integration of the interventions within the curriculum and with relevant educative outcomes could mitigate common reasons for implementation failure such as time constraints, competing curricular demands and overburdened teachers [19, 110].

## Strengths and limitations

This is the first systematic review to comprehensively synthesise the effects of school-based HIIT interventions across a wide range of health and wellbeing outcomes. The review has conducted a rigorous assessment of the risk of bias of included studies and available evidence, which allows the results to be interpreted with the required caution. Further, the review includes several meta-analyses and subsequent meta-regressions, which provide novel insights into the impact of HIIT in this setting along with associated factors. A limitation of this review includes the potential publication bias from only using articles published in English and omitting literature that was not peer-reviewed. Additionally, the papers included within this systematic review were mainly studies with small sample sizes and were classified as having a high risk of bias. Therefore, the results may need to be interpreted with caution.

## Conclusion

HIIT is an effective strategy for improving various health outcomes within the school setting, with our meta-analyses indicating meaningful improvements in markers of body size and composition, cardiovascular disease blood biomarkers, and CRF when compared to a non-exercise control group. However, our risk of bias results highlight that more high-quality studies are needed in this area. Currently, there is insufficient evidence to suggest that HIIT is superior to moderate continuous exercise or other types of comparative exercise. It is recommended that future research addresses the paucity of information on cognitive, physical activity, and nutrition outcomes associated with school-based HIIT interventions. It is also recommended that future research examines the effectiveness of these interventions over longer periods and how the interventions can be best developed and integrated within school practice to ensure engagement and maintenance.

## Supporting information

**S1 File. Search terms.**
(DOCX)

**S2 File. Certainty of evidence based on Grading of Recommendations, Assessment, Development and Evaluation (GRADE).**
(DOCX)

**S3 File. Body composition forest plots.**
(PDF)

**S1 Dataset.**
(XLSX)

**S1 Checklist. PRISMA checklist.**
(DOCX)

## Acknowledgments

The authors would like to thank Kate Sansum and Jenna Markey, University of Exeter students, and Tracy Bruce, University of Queensland academic librarian, for their assistance and contributions to this review through the search and data extraction phases.

## Author Contributions

**Conceptualization:** Alan R. Barker, Bert Bond, Renae Earle, Jo Varley-Campbell, Dimitris Vlachopoulos, Jacqueline L. Walker, Kathryn L. Weston, Michalis Stylianou.

**Data curation:** Stephanie L. Duncombe, Renae Earle.

**Formal analysis:** Stephanie L. Duncombe, Jo Varley-Campbell.

**Methodology:** Stephanie L. Duncombe.

**Supervision:** Alan R. Barker, Michalis Stylianou.

**Visualization:** Stephanie L. Duncombe.

**Writing – original draft:** Stephanie L. Duncombe.

**Writing – review & editing:** Stephanie L. Duncombe, Alan R. Barker, Bert Bond, Renae Earle, Jo Varley-Campbell, Dimitris Vlachopoulos, Jacqueline L. Walker, Kathryn L. Weston, Michalis Stylianou.

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
