## [Decision Letter · Decision Letter 0]

10 Feb 2022

PONE-D-22-01744School-based high-intensity interval training programs in children and adolescents: A Systematic Review and Meta-AnalysisPLOS ONE

Dear Dr. Duncombe,

Thank you for submitting your manuscript to PLOS ONE. After careful consideration, we feel that it has merit but does not fully meet PLOS ONE’s publication criteria as it currently stands. Therefore, we invite you to submit a revised version of the manuscript that addresses the points raised during the review process. Overall, the your manuscript was well-received by reviewers. Each noted minor issues to address, please read each set carefully. In light of this, I reviewed the paper myself and found it to be well-written and thorough. Work like this is helpful for future research designs, so thank you for your contributions. 

We look forward to receiving your revised manuscript.

Kind regards,

Chris Harnish, PhD

Academic Editor

PLOS ONE

Journal Requirements:

Reviewers' comments:

Reviewer's Responses to Questions

**Comments to the Author**

1. Is the manuscript technically sound, and do the data support the conclusions?

Reviewer #1: Yes

Reviewer #2: Yes

Reviewer #3: Yes

2. Has the statistical analysis been performed appropriately and rigorously? 

Reviewer #1: Yes

Reviewer #2: Yes

Reviewer #3: I Don't Know

3. Have the authors made all data underlying the findings in their manuscript fully available?

Reviewer #1: Yes

Reviewer #2: Yes

Reviewer #3: No

4. Is the manuscript presented in an intelligible fashion and written in standard English?

Reviewer #1: Yes

Reviewer #2: Yes

Reviewer #3: Yes

5. Review Comments to the Author

Reviewer #1: General Comments

This study performed a systematic review and meta-analysis on the physiological effects of high intensity interventions on school aged children. This is an important and needed study. The authors should be commended for using PRISMA and a priori registering this protocol. Overall this is a well designed, performed, and written study. Please see minor points below.

Introduction

Overall the introduction reads well. I commend the authors for including previous systematic reviews and meta-analyses and why this specific SR and meta needs to be performed. This provides greater transparency and an improved foundational argument for this specific study.

Methods

I commend the authors for using PRISMA and a priori registering this study. I also commend the authors for including the full search strategy as an appendix for reproducibility, and the use of a medical librarian.

Can you clarify why quasi-experimental studies were included?

Suggest excluding fixed effects reporting as this may confuse the reader.

I commend the authors for the overall well designed meta-analyses and meta-regression.

Results

The risk of bias table is a bit busy and hard to read. Suggest splitting the table into A.B.C. tables for different RoB tools.

Suggest adding at least one forest plot to the main results to improve readability and interpretation for the lay reader.

Discussion

Overall the main summary is well written.

Reviewer #2: The authors present a very interesting review of the works concerning HIIT applied in school age.

I absolutely agree with the conclusions as this scheme could be more effective and pleasing to the kids.

In my opinion some aspects should be underlined:

- divide the studies into two age groups 5-12 and 12-17 as due to puberty the response to exercise and body composition are different

- also consider the difference in sex

- it would be interesting to have a table with a summary of the most frequently used schemes, perhaps suggesting some that can be used, both in practice and in future studies.

- a correlation with the countries where the studies were carried out would also be interesting since, unfortunately, the practice and attention to physical activity is not uniform.

- finally, a stratification based on the assessment of body composition would be interesting, which may go beyond the BMI (for example plicometry or bioimpedance)

Reviewer #3: You have provided a comprehensive review of the included papers, and I commend your analysis techniques.

As you have declared that a large majority of the selected papers show a high risk of bias (line 202) this should be reflected in your conclusion (line 448 onwards), and you may feel it appropriate to make recommendations for future studies (line 453 onwards) to overcome these issues of bias.

Please also explain the exceptional situation that restricts data availability.

6. PLOS authors have the option to publish the peer review history of their article (what does this mean?). If published, this will include your full peer review and any attached files.

Reviewer #1: **Yes: **Garrett Bullock

Reviewer #2: **Yes: **Roberto Cannataro

Reviewer #3: No

---

## [Author Response · Author response to Decision Letter 0]

28 Feb 2022

We thank the reviewers for their valuable time and useful feedback. All our responses our outlined in our "Response to Reviewers".

---

## [Decision Letter · Decision Letter 1]

21 Mar 2022

School-based high-intensity interval training programs in children and adolescents: A Systematic Review and Meta-Analysis

PONE-D-22-01744R1

Dear Dr. Duncombe,

We’re pleased to inform you that your manuscript has been judged scientifically suitable for publication and will be formally accepted for publication once it meets all outstanding technical requirements. I wish to thank the authors on behalf of the reviewers who spoke highly of the paper and appreciated the added work to make it better.

Kind regards,

Chris Harnish, PhD

Academic Editor

PLOS ONE

Additional Editor Comments (optional):

Reviewers' comments:

Reviewer's Responses to Questions

**Comments to the Author**

1. If the authors have adequately addressed your comments raised in a previous round of review and you feel that this manuscript is now acceptable for publication, you may indicate that here to bypass the “Comments to the Author” section, enter your conflict of interest statement in the “Confidential to Editor” section, and submit your "Accept" recommendation.

Reviewer #1: All comments have been addressed

Reviewer #2: All comments have been addressed

2. Is the manuscript technically sound, and do the data support the conclusions?

Reviewer #1: Yes

Reviewer #2: Yes

3. Has the statistical analysis been performed appropriately and rigorously? 

Reviewer #1: Yes

Reviewer #2: Yes

4. Have the authors made all data underlying the findings in their manuscript fully available?

Reviewer #1: Yes

Reviewer #2: Yes

5. Is the manuscript presented in an intelligible fashion and written in standard English?

Reviewer #1: Yes

Reviewer #2: Yes

6. Review Comments to the Author

Reviewer #1: The authors should be commended for a well performed and written study. The authors have appropriately and thoroughly responded to the original reviewer comments. There are no more edits, well done.

Reviewer #2: I think the manuscript was already written in a good shape, the authors have improved it further so I think it is suitable for publication.

I hope that this work can be an impulse for the use of this technique also in school groups, not only for high level athletes.

7. PLOS authors have the option to publish the peer review history of their article (what does this mean?). If published, this will include your full peer review and any attached files.

Reviewer #1: **Yes: **Garrett Bullock

Reviewer #2: **Yes: **Roberto Cannataro

---

## [Editor Report · Acceptance letter]

24 Mar 2022

PONE-D-22-01744R1 

School-based high-intensity interval training programs in children and adolescents: A Systematic Review and Meta-Analysis 

Dear Dr. Duncombe:

I'm pleased to inform you that your manuscript has been deemed suitable for publication in PLOS ONE. Congratulations! Your manuscript is now with our production department. 

Kind regards, 

on behalf of

Dr. Chris Harnish 

Academic Editor

PLOS ONE